# Clinical Usefulness of Non-Invasive Metabolic-Associated Fatty Liver Disease Risk Assessment Methods in Patients with Full-Blown Polycystic Ovary Syndrome in Relation to the MRI Examination with the Ideal IQ Sequence

**DOI:** 10.3390/biomedicines10092193

**Published:** 2022-09-05

**Authors:** Łukasz Blukacz, Artur Nowak, Mariusz Wójtowicz, Angelika Krawczyk, Grzegorz Franik, Paweł Madej, Dagmara Pluta, Karolina Kowalczyk, Michał Żorniak

**Affiliations:** 1Department of Gynecological Endocrinology, School of Medicine in Katowice, Medical University of Silesia, 40-752 Katowice, Poland; 2Gynecological and Obstetrician Polyclinic, 15-435 Białystok, Poland; 3Department of Gynecological and Obstetrics, Women’s and Child Health Center, Medical University of Silesia, 41-803 Zabrze, Poland; 4Student Scientific Association of Gynecological Endocrinology, School of Medicine in Katowice, Medical University of Silesia, 40-752 Katowice, Poland; 5Department of Gastroenterological Oncology, The Maria Sklodowska-Curie National Institute of Oncology, 02-781 Warsaw, Poland; 6Department of Gastroenterology, Hepatology and Clinical Oncology, Centre of Postgraduate Medical Education, 01-813 Warsaw, Poland

**Keywords:** polycystic ovary syndrome, metabolic associated fatty liver disease, lipid accumulation product, free androgen index

## Abstract

The coexistence of polycystic ovary syndrome (PCOS) and liver steatosis has been studied for years. The gold standards for the diagnosis of liver steatosis are liver biopsy and magnetic resonance imaging (MRI), which are invasive and expensive methods. The main aim of this study is to check the usefulness of lipid accumulation product (LAP) and free androgen index (FAI) in the diagnosis of liver steatosis. The Ideal IQ MRI was performed in 49 women with PCOS phenotype A to assess the degree of liver steatosis, which was expressed with the proton density fat fraction (PDFF). Anthropometric examination and laboratory tests were performed, and the LAP and FAI were calculated. The correlation between MRI results and LAP, FAI, and one of the FAI components, sex hormone binding globulin (SHBG), was checked using statistical tests. There is a statistically significant correlation between PDFF and LAP and also between PDFF and FAI. LAP = 70.25 and FAI = 5.05 were established as cut-offs to diagnose liver steatosis. The SHBG is not a statistically significant parameter to predict liver steatosis. The study showed that especially LAP, but also FAI, can be used to predict liver steatosis with high specificity and sensitivity.

## 1. Introduction

Polycystic ovary syndrome (PCOS) is a common gynecological condition with a prevalence of approximately 21% of women of reproductive age [1]. Current diagnostic criteria are the Rotterdam criteria containing ovulatory dysfunction such as oligo and anovulation, polycystic ovarian morphology (PCOM) in ultrasound examination, and clinical or biochemical hyperandrogenism [1]. The PCOM is defined as 12 or more follicles of 2–9 mm and/or an ovarian volume of more than 10 mL [2]. Clinical hyperandrogenism is defined as hirsutism, androgenetic alopecia, and acne [3]. Hirsutism is a situation in which the terminal, dark, coarse hairs appear in females in male-like distributions, such as the upper lip, chin, chest, upper and lower back, upper and lower abdomen, upper arms, and thighs.

Total testosterone, free testosterone, androstenedione, dehydroepiandrosterone sulfate, and 17-OH-progesterone are measured in the follicular phase to assess biochemical hyperandrogenemia [3].

The manifestations of ovulatory dysfunction are oligomenorrhea, secondary amenorrhea, and difficulties with getting pregnant [4]. Oligomenorrhea means fewer than nine menstruations in a year or three or more cycles lasting longer than 35–38 days in a year. There are four PCOS phenotypes (A–D), depending on the criteria presented by the patient. In phenotype A, three symptoms from Rotterdam criteria need to occur [1]. Phenotype B means ovulatory dysfunction and hyperandrogenism. Phenotype C means hyperandrogenism and PCOM, and phenotype D means ovulatory dysfunction and PCOM. To diagnose PCOS, other causes of hyperandrogenism and ovulation disorders such as hyperprolactinemia, thyroid disorders, hypercortisolemia, non-classic congenital adrenal hyperplasia, premature ovarian failure, acromegaly, and malignancies should be ruled out.

Obesity is a very common problem among women with PCOS. It is known that increasing the volume of adipocytes induces changes in the metabolism of adipose tissue and consequently chronic inflammation and the development of insulin resistance [5,6,7]. Mainly visceral adipose tissue is responsible for the production of pro-inflammatory cytokines, which are released as a result of adipocyte growth and hypoxia [7].

According to the definition of the World Health Organization (WHO), obesity means a body mass index (BMI) higher than 30 kg/m^2^ [8]. This definition does not include proportions of subcutaneous adipose tissue and visceral adipose tissue that are significant to estimate metabolic risk [9]. It seems to be more useful to use a division into four categories: normal-weight obese, metabolically obese normal-weight, metabolically healthy obese, and metabolically unhealthy obese [10]. The first two groups include patients with BMI below 25 kg/m^2^ but incorrect lipid profile and glucose metabolism [10]. To measure visceral adipose tissue, the lipid accumulation product (LAP) can be used. It is necessary to know waist circumference (WC) and triglycerides (TG) concentration to calculate LAP according to the formula:LAP for women = (WC [cm] − 58) × (TG concentration [mmol/L]).

LAP can be used for estimating cardiovascular risk [11], diagnosing metabolic syndrome [11], and diagnosing hormonal disorders and metabolic complications in women with PCOS [12], but also for diagnosing non-alcoholic fatty liver disease (NAFLD) [13]. The division into visceral and subcutaneous adipose tissue is not the only division. We can also distinguish white and brown adipose tissue, but this distinction is not metabolically significant. Not only obesity, but overweight, which means a BMI between 25 and 30, also has a negative effect on health.

Insulin resistance, which affects most PCOS women, causes the development of hyperinsulinemia. Insulin stimulates the synthesis of androgens in the ovaries, which results in hyperandrogenism and, consequently, a decrease in the level of adiponectin. It is a positive factor in lipid metabolism, and lower levels of it can cause liver steatosis. Due to the described mechanism, the coexistence of PCOS with liver diseases resulting from fat accumulation is possible [14]. Non-alcoholic fatty liver disease (NAFLD) is a chronic liver disease caused by steatosis of hepatocytes that is not associated with alcohol consumption [15]. Recently, a new name was proposed, metabolic-associated fatty liver disease (MAFLD), and this new name is preferred [16]. The reason for this decision is a very strong association of the disease with metabolic syndrome. This new name will be used below instead of NAFLD. To be diagnosed with MAFLD, patients need to present two or more of the following signs of metabolic syndrome: waist circumference ≥ 80 cm for women and ≥94 cm for men, blood pressure ≥ 130/85 mmHg or hypertension therapy, fasting glucose ≥ 100 mg/dL or antidiabetic therapy, TG > 150 mg/dL or hypolipidemic therapy, and HDL < 50 mg/dL for women and <40 mg/dL for men [17]. The most important criterion is steatosis of hepatocytes, which can be recognized by liver biopsy [18]. According to the previous name of the disease, in order to make a diagnosis, regular alcohol consumption needs to be excluded. There are a few non-invasive methods to diagnose steatosis, such as ultrasound examination (USG), computed tomography (CT), or magnetic resonance imaging (MRI). Transient elastography (TE) performed with ultrasonography is the most recommended non-invasive method [16]. PCOS, hypothyroidism, genetic disorders, some drugs, and many other things can be the causes of MAFLD, even among patients with a proper BMI [19]. Drugs that need special attention, as they can cause liver steatosis, include steroids, methotrexate, tamoxifen, amiodarone, and antidepressants [19]. Conditions such as hemochromatosis, Wilson’s disease, autoimmune enteritis, viral hepatitis, and α1-antytripsin deficiency also need to be excluded. Another indicator that can be used to predict liver fibrosis is fibrosis-4 (FIB-4). This requires knowledge of age, alanine aminotransferase, aspartate aminotransferase, and platelets. It is a non-invasive indicator [20].

## 2. Materials and Methods

Fifty-four patients of the Department of Gynecological Endocrinology, Medical University of Silesia in Katowice in Poland, in 2015–2016, aged 18–39, suffering from PCOS phenotype A, were classified for the study. Five of them did not consent to Ideal IQ sequence MRI, and the final number of study participants was forty-nine.

The exclusion criteria were hyperprolactinemia, hypercortisolemia, thyroid disorders, hormonal contraceptive therapy, steroids intake, antiandrogens intake, dietary supplements intake in the last three months, viral and autoimmune liver diseases, focal changes in the liver, and alcohol consumption over 20 g per day. Patients denied any fluctuations of body weight in the last six months.

This prospective observational study aims to check the usefulness of lipid accumulation product (LAP) and free androgen index (FAI) in diagnosing liver steatosis in 49 women with PCOS phenotype A.

A gynecological interview was conducted, including information about menstrual cycles, obstetric history, and hormonal disorders. Patients were asked about taking supplements and drugs. The patients’ medical documentation was analyzed, if available. Transvaginal ultrasound examination was performed by the same doctor using Voluson 730 Expert, the PCOM was confirmed according to Rotterdam ESHRE/ASRM-Sponsored PCOS Consensus Workshop Group 3, and other causes of liver damage were excluded. The patients were assessed on the Ferriman–Gallwey Scale, the Global Acne Severity Scale, and the Ludwig Scale. A Ferriman–Gallwey score of 8 or more means a diagnosis of hirsutism. A score of 0 points on the Global Acne Severity Scale means clear skin. A score of 1 or more points was awarded to patients with acne, and the classification was as follows: 1—almost clear skin, 2—mild severity, 3—moderate severity, 4—severe, 5—very severe. The Ludwig Scale was used to assess androgenetic alopecia. According to that scale, patients with baldness were classified into one of three groups: Type I—mild, type II—moderate, and type III—extreme hair loss. The body weight, height, and blood pressure were measured. The waist circumference was measured at the halfway point between the iliac crest and the lower point of the last rib in the horizontal plane passing through the navel without compressing the skin. The hip circumference was measured at the largest circumference around the buttocks. Before the measurements, the patients were without meals for 8 h.

Laboratory tests were performed using samples of blood taken in the morning, after 12 h without meals, between the second and fifth days of the menstrual cycle. Estradiol, follicular stimulating hormone (FSH), luteinizing hormone (LH), total and free testosterone, 17-OH-progesterone, androstenedione, cortisol, dehydroepiandrosterone sulfate (DHEAS), sex hormone binding globulin (SHBG), prolactin, and insulin were determined by enzyme-linked immunosorbent assay (ELISA) using DRG Diagnostics GmbH reagents. Glucose profile and lipid profile, including total cholesterol, HDL, and LDL fractions, were determined by colorimetric method using AU 680 analyzer and Beckman Coulter reagents. Additionally, a complete blood count was performed.

A 2-point oral glucose tolerance test (OGTT), using 75 g of glucose, was performed.

An assay for the qualitative determination of HBsAg and HCV antibodies was also performed. Patients with positive results of these tests were excluded from the study and referred for further diagnosis of viral hepatitis.

Homeostatic Model Assessment—Insulin Resistance (HOMA-IR) was used according to the formula:fasting blood glucose (mmol/L) × fasting insulin (mU/L)/22.5

A score of more than 2 confirmed insulin resistance [21].

The free androgen index (FAI) was calculated using the following formula [22]:total testosterone (nmol/mL) × 100%/SHBG (nmol/mL)
and the following formula
(WC [cm] − 58) × (TG concentration [mmol/L])
was used to calculate LAP.

Magnetic resonance spectroscopy was used to assess the degree of liver steatosis. It is a non-invasive method that does not use X-rays but uses the movement of water molecules to image organs. The Dixon technique, which is known as the chemical shift technique, was used. Due to the difference in the signal of water and triglycerides, a picture of fatty liver can be obtained.

A GE Healthcare Discovery MR750 3T MRI scanner was used for the quantitative assessment of liver steatosis in an Ideal IQ sequential. It is the most modern method used for this purpose. Examinations were performed in District Hospital of Orthopedics and Trauma Surgery Piekary Slaskie, Poland, with the consent of the Bioethics Committee.

The examination was performed in the supine position while patients held their breath, for no longer than 25 s, and it covered the entire liver. Water and fat images were reconstructed, and a fraction map was obtained. All confounding factors have been taken into account. In the next step, special Ideal IQ software dedicated on AW 4.6 GE Healthcare workstation, was used to analyze data. Regions of interest were 2 cm^2^ in area, oval in shape, and contained no hepatic vessels or motor artifacts. They were placed on three different cross sections of proton density fat fraction (PDFF) maps.

Liver steatosis was expressed by PDFF, and according to the current recommendations, fatty liver disease was confirmed for the results >5.56% [20].

The calculations were performed using STATA/SE 14.2—StataCorp. 2015. Stata Statistical Software: Release 14. College Station, TX: StataCorp LP.

Descriptive statistics were performed.

A Spearman correlation coefficient (rho) was used to measure the strength of association between PDFF and individual variables.

Receiver Operating Characteristic (ROC) curve, sensitivity, specificity, positive predictive value (PPV), and negative predictive value (NPV) were used to determine the values that optimally differentiate the level of liver steatosis. Logistic regression was used to obtain odds ratio (OR) of being in the PDFF > 5.56% group depending on analyzed variables.

The test results were considered statistically significant at *p* < 0.05.

## 3. Results

### 3.1. Descriptive Statistics

Descriptive statistics are presented in Table 1. Data deficiencies in blood pressure, blood morphology, GGTP, Fe, OGTT, G/I, HOMA IR, free testosterone, androstenedione, result from the retrospective nature of the study and the inability to complete them.

### 3.2. Correlation Analysis

In 22 of 49 patients, the MAFLD was confirmed by the Ideal IQ MRI.

Statistically significant correlations between PDFF and individual variables were found, including negative significant correlations between PDFF and glucose-to-insulin ratio (G/I) and SHBG (Table 2). Other variables were non-significantly correlated with PDFF.

### 3.3. Assessment of LAP Usefulness

The ROC curve to determine the cut-off point of LAP for PDFF > 5.56% was drawn (Figure 1). LAP = 70.25 was established as a cut-off. The area under the receiver operating characteristic (AUROC) was 0.83. LAP = 70.25 is statistically significant for prediction liver steatosis for PDFF > 5.56%.

The results showed that 19 of 26 patients with LAP > 70.25 had liver steatosis and 3 patients with liver steatosis had LAP ≤ 70.25 (Table 3).

The parameters for being in the group PDFF > 5.56 and LAP > 70.25: OR = 18.10 (95% C.I. 4.07; 80.38), chi2(1) = 17.78, *p* < 0.001, prediction sensitivity 86.36%, prediction specificity 74.07%, PPV = 73.08% and NPV = 86.96%.

### 3.4. Assessment of FAI Usefulness

The ROC curve to determine the cut-off point of FAI for PDFF > 5.56% was drawn (Figure 2). FAI = 5.05 was established as a cut-off. The area under the receiver operating characteristic (AUROC) was 0.77. FAI = 5.05 was statistically significant to prediction liver steatosis for PDFF > 5.56%.

The results showed that 16 of 22 patients with FAI > 5.05 had liver steatosis, and 6 patients with liver steatosis had FAI ≤ 5.05 (Table 4).

The parameters for being in the group PDFF > 5.56 and FAI > 5.05: OR = 11.73 (95% C.I. 3.04; 45.27), chi2(1) = 14.55, *p* < 0.001, prediction sensitivity 72.73%, prediction specificity 81.48%, PPV = 76.19% and NPV = 78.57%.

### 3.5. Assessment of SHBG Usefulness

The ROC curve to determine the cut-off point of SHBG, analyzed as a component of FAI, for PDFF > 5.56% was drawn (Figure 3). SHBG = 42.35 was established as a cut-off. The area under the receiver operating characteristic (AUROC) was 0.79. SHBG = 42.35 allows prediction of liver steatosis for PDFF > 5.56%.

We found that 4 of 16 patients with SHBG > 42.35 had liver steatosis (Table 5).

The parameters for being in the group PDFF > 5.56 and SHBG > 42.35: OR = 0.28 (95% C.I. 0.07; 1.04), chi2(1) = 3.80, *p* = 0.051 and it is not statistically significant, prediction sensitivity 18.18%, prediction specificity 56.55%, PPV = 25.00%, and NPV = 45.45%.

## 4. Discussion

The present work confirms that the LAP is an indicator of liver steatosis with high predictive value. According to the results, there is a 95% confidence interval that women with LAP > 70.25 have a 4.07 to 80.38 times higher risk of liver steatosis.

Using the MRI method, which is the gold standard, like liver biopsy, in diagnosing steatosis, made the results more reliable. Another statistically significant parameter in the prediction of liver steatosis, according to the study, is FAI. There is a 95% confidence interval that women with FAI.5.05 have a 3.04 to 45.27 higher risk of liver steatosis. SHBG, as a component of FAI, was not a point of interest in this study. During the analysis, the negative correlation between it and PDFF made it interesting; as a result, the usefulness of SHBG in MAFLD risk assessment in women with PCOS was checked. SHBG is not a statistically significant parameter in the prediction of liver steatosis. The results of the study show that non-invasive and inexpensive parameters can be used to diagnose liver steatosis. It is important for women suffering from PCOS because the prevalence of MAFLD in this group is approximately 30–70% [23]. For comparison, in the control group, it is 20–30% [23]. Moreover, the prevalence of MAFLD is approximately two times higher in the group of patients with phenotype A than in the others [24]. The prevalence of MAFLD in the study group was 44.9%, and in 22 of 49 patients, the diagnosis was confirmed. This is consistent with the published data [23]. We found that 71.4% of the analyzed group were women with BMI > 30. Indeed, obesity increases the risk of MAFLD, but the disease is also possible in patients with proper BMI [19].

In both PCOS and MAFLD, there are glucose and lipid disorders and hypertension. Insulin resistance is diagnosed in most PCOS patients but also in most MAFLD patients, regardless of the BMI value. It needs to be emphasized that this factor is significant in both disorders’ pathophysiology and can be an explanation of their coexistence [23]. Increased insulin levels induce androgen synthesis and inhibit SHBG synthesis. The result is hyperandrogenemia, which decreases adiponectin levels and may be one of the causes of liver steatosis [14]. Because MAFLD is diagnosed in many patients when complications occur, it is reasonable to identify risk groups and diagnose them. There are some published studies about diagnosing MAFLD, especially in the PCOS group. Most of them used ultrasound examination as a diagnostic method [25,26,27]. It is known that USG is not a preferable method to diagnose liver steatosis. In 2006, an analysis of 200 PCOS patients was published. In six of them, biopsy confirmed nonalcoholic steatohepatitis (NASH) [28]. This was the beginning of the interest in the coexistence of PCOS and liver steatosis. Since then, every few years, publications have been appearing on the subject from various sides. Few of the published studies used liver biopsy to confirm steatosis [28,29,30]. Even though it is a gold-standard method, it is also an invasive method and cannot be used as a screening test. Interestingly, laboratory tests such as alanine aminotransferase (AlAT) are higher in approximately 50% of MAFLD patients, so it is excluded from the risk group assessment [19].

There are few published studies about using LAP in diagnosing MAFLD, but in all of them, MAFLD was confirmed by laboratory tests and USG [31,32].

The present study uses Ideal IQ MRI as a confirmation method. Of course, the costs of this method exclude it as a screening method, but it allowed us to obtain results with high sensitivity and specificity. The costs of the MRI examination were the limiting factor of the study, and because of that, the studied group was quite small. Still, the study’s results show that LAP and FAI should be used in clinical practice.

## 5. Conclusions

The lipid accumulation product > 70.25 and the free androgens index > 5.05 correlate with PDFF > 5.56%. They can be used in screening diagnosis of MAFLD before liver biopsy and MRI in women with PCOS. It can reduce costs, invasiveness, and it will especially increase the detection of the disease before complications occur. PCOS is only one of the risk groups of MAFLD, and this is the subject of this study, but there are lots of different risk groups in which study results could be useful. The main conclusion is that LAP and FAI are non-invasive methods of screening liver steatosis and can be used in various ways in clinical practice.

## Figures and Tables

**Figure 1 biomedicines-10-02193-f001:**
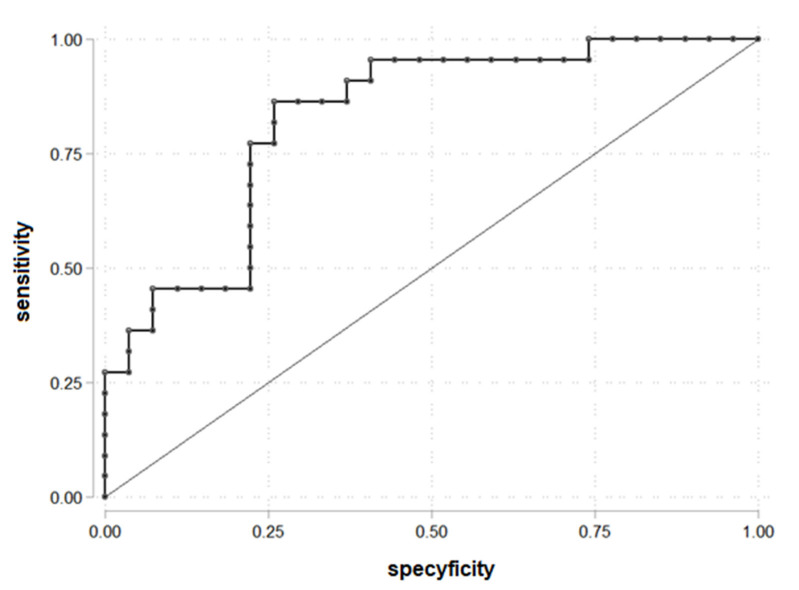
ROC curve for prediction PDFF > 5.56% according to the LAP (*n* = 49).

**Figure 2 biomedicines-10-02193-f002:**
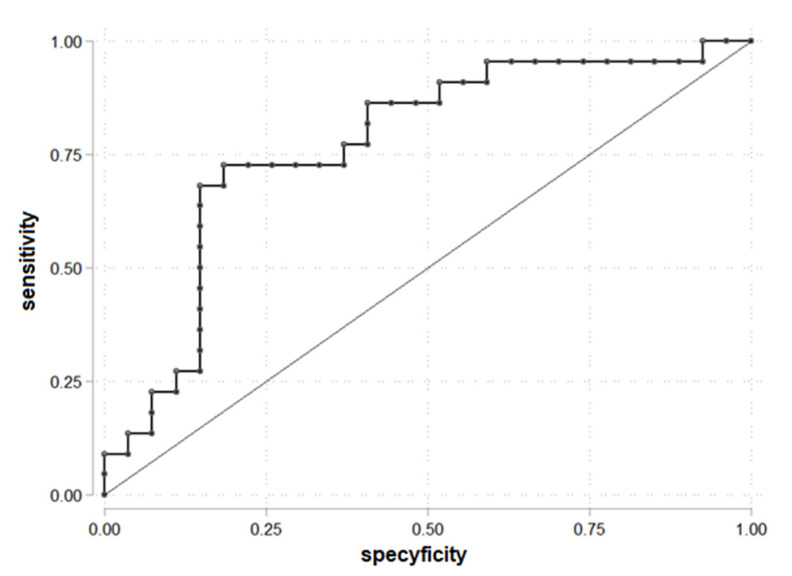
ROC curve for prediction PDFF > 5.56% according to the FAI (*n* = 49).

**Figure 3 biomedicines-10-02193-f003:**
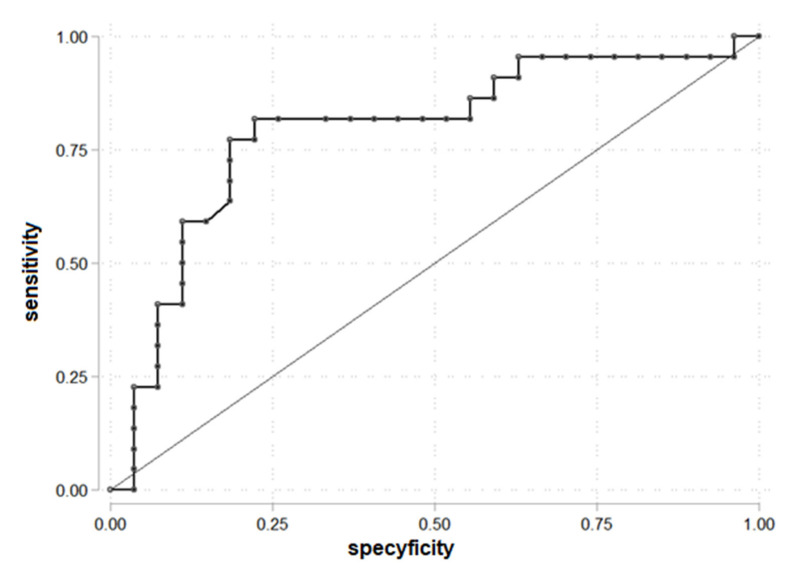
ROC curve for prediction PDFF > 5.56% according to the SHBG (*n* = 49).

**Table 1 biomedicines-10-02193-t001:** Descriptive statistics of the studied group.

Variable (Unit)	*n*	Mean ± Standard Deviations	Minimum	Maximum
Age	49	26.04 ± 5.22	18	39
Heigh (cm)	49	164.02 ± 6.35	149	177
Weigh (kg)	49	83.24 ± 20.58	45	138
BMI (kg/m^2^)	49	30.93 ± 7.35	17.56	51.86
Waist (cm)	49	97.58 ± 18.91	62.00	134.00
Hips (cm)	49	111.86 ± 13.50	89.00	146.30
Waist to hips ratio	49	0.87 ± 0.11	0.65	1.12
Systolic blood pressure (mmHg)	47	123.13 ± 14.86	90	160
Diastolic blood pressure (mmHg)	47	80.55 ± 12.53	50	100
White blood cells (10^3^/µL)	43	6.73 ± 2.02	3.43	13.00
Red blood cells (10^6^/µL)	43	4.62 ± 0.34	3.53	5.18
Hemoglobin (g/dL)	43	13.73 ± 0.96	11.10	15.70
Platelets (10^3^/µL)	43	304.95 ± 189.98	145	1462
TG (mg/dL)	49	162.39 ± 70.95	71	382
TC (mg/dL)	49	200.71 ± 40.54	39	292
HDL (mg/dL)	49	57.29 ± 16.49	37	122
LDL (mg/dL)	49	116.43 ± 33.08	51	209
ALAT (U/L)	49	36.18 ± 24.75	11	148
ASPT (U/L)	49	29.22 ± 14.74	12	101
GGTP (U/L)	44	31.77 ± 24.53	10	125
Fe (µg/dL)	16	101.63 ± 27.70	45	153
GLU (mg/dL)	49	91.65 ± 9.69	68	119
OGTT (mg/dL)	48	112.71 ± 32.08	40	188
G/I	47	8.23 ± 7.06	1.52	48.02
HOMA IR	48	3.91 ± 3.19	0.48	15.60
Free testosterone (pg/mL)	48	4.02 ± 3.04	0.29	14.41
Total testosterone (ng/dL)	49	0.48 ± 0.18	0.10	0.90
DHEAS (µg/dL)	49	383.99 ± 130.36	117.20	700.60
Androstenedione (ng/mL)	48	3.41 ± 1.79	0.82	9.51
17-OH-progesterone (ng/mL)	49	2.12 ± 1.17	0.90	8.43
SHBG (ng/dL)	49	40.43 ± 25.57	13.20	156.00
FAI	49	5.42 ± 3.46	0.22	15.95
LAP	49	74.97 ± 46.68	4.43	198.15
PDFF	49	8.77 ± 8.63	1.96	40.50

**Table 2 biomedicines-10-02193-t002:** Statistically significant rho-Spearman coefficient between variables and PDFF.

	PDFF		
Variable (Unit)	rho	*p*	*n*
Weight (kg)	0.62	<0.0001	49
BMI (kg/m^2^)	0.67	<0.0001	49
Waist (cm)	0.59	<0.0001	49
Hips (cm)	0.50	0.0003	49
Waist to hips ratio	0.42	0.0030	49
Triglycerides (mg/dL)	0.30	0.0377	49
Glucose (mg/dL)	0.34	0.0165	49
OGTT	0.44	0.0020	48
Glucose/insulin	−0.50	0.0003	47
HOMA IR	0.55	<0.0001	48
SHBG (ng/mL)	−0.51	0.0002	49
FAI	0.54	0.0001	49
LAP	0.58	<0.0001	49

**Table 3 biomedicines-10-02193-t003:** The number of patients with liver steatosis depending on LAP.

	PDFF		
LAP	≤5.56	>5.56	Total
**≤70.25**	20	3	23
**>70.25**	7	19	26
**Total**	27	22	49

**Table 4 biomedicines-10-02193-t004:** The number of patients with liver steatosis depending on FAI.

	PDFF		
FAI	≤5.56	>5.56	Total
**≤5.05**	22	6	28
**>5.05**	5	16	21
**Total**	27	22	49

**Table 5 biomedicines-10-02193-t005:** The number of patients with liver steatosis depending on SHBG.

	PDFF		
SHBG	≤5.56	>5.56	Total
**≤42.35**	15	18	33
**>42.35**	12	4	16
**Total**	27	22	49

## Data Availability

Not applicable.

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
