# Peer review of "Clinical Usefulness of Non-Invasive Metabolic-Associated Fatty Liver Disease Risk Assessment Methods in Patients with Full-Blown Polycystic Ovary Syndrome in Relation to the MRI Examination with the Ideal IQ Sequence"

_biomedicines, 2022, doi:10.3390/biomedicines10092193_

Round 1

Reviewer 1 Report

The manuscript entitledClinical Usefulness of Non-Invasive Metabolic-Associated Fatty Liver Disease Risk Assessment Methods in Patients with Polycystic Ovary Syndrome in relation to the MRI examination with the Ideal IQ sequence described the usefulness of lipid accumulation product (LAP) and free androgen index (FAI) in the diagnosis of liver steatosis. It has a clear structure, rigorous logic, substantial content, and coherent sentences. This manuscript is of great significance in the primary screening of MAFLD in PCOS patients. Overall, the manuscript is well written, but there are several issues as described below:

1.The title is PCOS patients, while the article only describes patients with PCOS phenotype A, the title is suggested to be revised to be more accurate.

2.In the introduction section, need to describe the relationship between PCOS and MAFLD, suddenly jump from PCOS to MAFLD, lack of logic.

Author Response

Thank you very much for reading and analyzing our manuscript. As suggested in point 1, we changed the title. As suggested in point 2, a section on the relationship between PCOS and MAFLD has been added to the introduction. Please see the attachment. The revised version includes the suggestions of each reviewer.

Reviewer 2 Report

Polycystic ovary syndrome (PCOS) is a common gynaecological condition whose prevalence is approximately 21% of women in reproductive age.

Non-alcoholic fatty liver disease (NAFLD) is a chronic liver disease caused by steatosis of hepatocytes which is not associated with alcohol consumption. Recently, a new name was proposed, metabolic associated fatty liver disease (MAFLD), and it is preferred. PCOS can cause MAFLD, even in patients with proper BMI.

This prospective observational study aims to check the usefulness of lipid accumulation product (LAP) and free androgen index (FAI) in diagnosing liver steatosis in 49 women with PCOS phenotype A.  

PCOS was confirmed according to Rotterdam ESHRE/ASRM-Sponsored PCOS Consensus Workshop Group 3, and other causes of liver damage were excluded.

Waist and hip circumference and blood pressure were measured.

Laboratory tests were performed between the second and fifth days of the menstrual cycle: estradiol, follicular stimulating hormone (FSH), luteinizing hormone (LH), total and free testosterone, 17-OH-progesterone, androstenedione, cortisol, dehydroepiandrosterone sulfate (DHEAS), sex hormone binding globulin (SHBG), prolactin and insulin, glucose profile and lipid profile, complete blood count.

A 2- point oral glucose tolerance test was performed, and Homeostatic Model Assessment – Insulin Resistance (HOMA-IR) was used to confirm insulin resistance (if >2).

LAP and FAI were calculated.

The patients underwent MRI proton density fat fraction (PDFF) estimation.

The authors found that PDFF statistically correlated positively with weight, BMI, waistline, hips, lipids, glucose, insulin resistance, FAI, and LAP. LAP=70.25 and FAI=5.05 were found as cut-offs to diagnose liver steatosis. The SHBG was not a statistically significant parameter for predicting liver steatosis.

The introduction set the stage correctly.

The methods are well described; therefore, the study is reproducible.

The discussion is deep.

The Ethics Committee approved the study.

I found the article very interesting and promising because there is a need to find new scores that quickly allow the diagnosis of MAFLD, within everyone's reach, at a low cost.

Author Response

Thank you very much for reading and analyzing our manuscript. We tried to apply your suggestions. Where possible, we have reduced the text, because we had to take into account the minimum word requirements in the journal. Please see the attachment. The revised version includes the suggestions of each reviewer.
